🔓 | **Open Peer Review** | Antimicrobial Chemotherapy | Research Article

# The novel bacteriocin romsacin from *Staphylococcus haemolyticus* inhibits Gram-positive WHO priority pathogens

Runa Wolden,[1] Kirill V. Ovchinnikov,[2] Hermoine J. Venter,[1] Thomas F. Oftedal,[2] Dzung B. Diep,[2] Jorunn Pauline Cavanagh[1]

**ABSTRACT**  *Staphylococcus haemolyticus* is an increasingly relevant nosocomial pathogen. The combination of multi-drug resistance and ability to form biofilms makes *S. haemolyticus* infections difficult to treat. Bacteriocins are ribosomally synthesized antimicrobial peptides produced by bacteria to inhibit growth of often closely related bacteria. Due to differences in the modes of action between bacteriocins and antibiotics, bacteriocins are normally equally potent against antibiotic-resistant and antibiotic-sensitive strains. To find bacteriocins able to inhibit *S. haemolyticus* and related species, clinical and commensal *S. haemolyticus* isolates (*n* = 174) were assayed for bacteriocin production. One commensal isolate produced an antimicrobial substance inhibiting *S. haemolyticus* and *Staphylococcus aureus*. The substance had physicochemical properties that are characteristic of bacteriocins. Purification, whole-genome sequencing, and mass spectrometry identified the antimicrobial as a novel two-peptide lantibiotic, hereafter named romsacin. The bacteriocin was active against a broad range of Gram-positive bacteria, such as the World Health Organization priority pathogens *S. aureus* [methicillin-resistant *S. aureus* (MRSA)] and *Enterococcus faecium* [vancomycin-resistant *E. faecium* (VRE)]. Importantly, the bacteriocin also eradicated *S. haemolyticus*, *Staphylococcus epidermidis*, MRSA, and VRE biofilms.

**IMPORTANCE**  Bacteria produce bacteriocins to inhibit growth of other bacterial species. We have studied the antimicrobial activity of a new bacteriocin produced by the skin bacterium *S. haemolyticus*. The bacteriocin is effective against several types of Gram-positive bacteria, including highly virulent and antibiotic-resistant strains such as *Staphylococcus aureus* and *Enterococcus faecium*. Effective antimicrobials are important for the treatment of infections and the success of major surgery and chemotherapy. Bacteriocins can be part of the solution to the global concern of antimicrobial resistance.

**KEYWORDS**  *Staphylococcus haemolyticus*, bacteriocin, antimicrobial resistance, biofilm, AMR, lanthipeptides, lantibiotics, CoNS, romsacin, WHO priority pathogens

*S*taphylococcus haemolyticus frequently causes hospital-acquired infections, especially affecting immunocompromised patients with indwelling medical devices (1, 2). Clinical isolates of *S. haemolyticus* are often multi-drug resistant and consequently resistant to antibiotics normally used to treat staphylococcal infections (1, 2). *S. haemolyticus* is a coagulase-negative staphylococcus (CoNS). The closely related coagulase-positive *Staphylococcus aureus* colonizes human skin and mucous membranes and is often part of the normal bacterial flora. However, the bacterium is simultaneously one of the most frequent causes of bacterial infections (1). Methicillin-resistant *S. aureus* (MRSA) and vancomycin-resistant *S. aureus* are classified as global high priority pathogens by the World Health Organization (WHO) (1, 3–5). Vancomycin-resistant *Enterococcus faecium* (VRE) is another priority pathogen, where the acquisition of glycopeptide resistance genes and adaptation to the nosocomial setting have allowed

Address correspondence to Jorunn Pauline Cavanagh, pauline.cavanagh@uit.no.

Dzung B. Diep passed away in December 2022.

The authors declare no conflict of interest.

See the funding table on p. 19.

*[This article was published on 31 October 2023 with a typographical error in Fig. 8. The error was corrected in the current version, posted on 22 November 2023.]*

it to become a successful opportunistic pathogen (4–6). It is believed that current antibacterial agents, including agents in development, are insufficient to address the rising concern of antibiotic resistance (1). A promising alternative or supplement to antibiotics is bacteriocins.

Bacteriocins are ribosomally synthesized antimicrobial peptides produced by bacteria and typically kill closely related species. Bacteriocins can also be broad spectrum and often have a mechanism different from antibiotics (7–9). Bacteriocins are currently classified based on the presence or absence of post-translational modifications (10). Bacteriocins that are post-translationally modified belong to class I, while class II are unmodified (11–14). The lantibiotics which belong to class I are characterized by the presence of thioether cross-links between a cysteine and a dehydrated serine or threonine to form the unusual amino acids lanthionine and methyllanthionine, respectively (15). Lanthipeptide biosynthesis involves dehydration and cyclization modifications to a precursor peptide LanA, followed by proteolysis and export of the bioactive bacteriocin Lanα. Lanthipeptide gene clusters encode dedicated proteins for their biosynthesis, including LanM, which performs dehydration and cyclization, and LanT$_P$, which removes the leader sequence by proteolytic cleavage and exports it to the extracellular space (T$_P$: transporter and peptidase) (16). By convention, LanA liberated from its leader sequence is referred to as the pro-, core-, or mature peptide in unmodified form, although the leader is removed after the peptide is modified (5, 17). Lantibiotic producers are immune to their own bacteriocin due to the production of immunity proteins (LanI) and/or ABC transporter proteins with immunity function (LanFE/LanFEG) (4, 18). Some lantibiotics are two-peptide bacteriocins consisting of Lanα and Lanβ, derived from LanA1 and LanA2 precursor peptides, which act synergistically to exert maximal antimicrobial activity (4–6).

In this study, we investigated 174 clinical and commensal *S. haemolyticus* isolates for bacteriocin production. The aim was to find new bacteriocins able to inhibit *S. haemolyticus* and related organisms, such as *S. aureus*. One commensal isolate inhibited both species. We discovered that the genome [previously sequenced in reference (11)] of this isolate contained a lanthipeptide biosynthetic gene cluster predicted to encode a new two-peptide lantibiotic. In this work, we describe the purification and characterization of the identified two-peptide lantibiotic. The bacteriocin was active against many Gram-positive bacteria such as VRE, MRSA, and *S. haemolyticus*. In addition, the bacteriocin eradicated *S. haemolyticus*, *Staphylococcus epidermidis*, *S. aureus*, and *E. faecium* biofilms.

## RESULTS

### *S. haemolyticus* produces bacteriocins

From the collection of 174 *S. haemolyticus* isolates, overnight cultures were spotted on lawns of a clinical isolate of *S. haemolyticus* and *Staphylococcus aureus*. *Lactococcus lactis* was also included as an indicator due to its broad and high sensitivity toward many bacteriocins. Growth inhibition (clear zone) against indicators was observed from three of the isolates (*S. haemolyticus* 53-34, 57-27, and 58-57). Cell-free supernatants were tested, and only *S. haemolyticus* 57-27 produced an antimicrobial that was temperature stable (4°C–121°C). It was also stable to pH (2–12) but protease sensitive (trypsin), which are all characteristics of bacteriocins. *S. haemolyticus* 57-27 was isolated from the groin of an asymptomatic carrier (11, 19).

### Lantibiotic genes found in *S. haemolyticus*

Assembled genomes (contigs) from 174 *S. haemolyticus* isolates were submitted to the BAGEL4 webserver to identify bacteriocin-encoding genes (20). Predicted bacteriocin gene clusters were found in all three genomes from *S. haemolyticus* isolates with antimicrobial activity. Two of the three isolates (isolate 58-57 and 53-34) were found to encode heat-labile (molecular weight >10 kDa) bacteriocins and was thus not investigated further. The remaining isolate (57-27) exhibiting inhibition contained a gene cluster

with homology to lantibiotic biosynthetic clusters. Two bacteriocin structural genes were predicted to encode the α- and β-components of a two-peptide lantibiotic with sequence homology to the A1 and A2 peptides of plantaricin W (Uniprot: D2KR94, Q9AF68). However, the two predicted core peptides shared only 67% and 51% identity to the A1 and A2 core peptides of plantaricin W, respectively. The relatively low sequence identity to known lantibiotics suggested that the cluster may encode a novel two-peptide lantibiotic (Table 1; Fig. 1). The gene product of *lanA2* is a class II lanthipeptide of the LchA2/BrtA2 family. This lanthipeptide was also uncovered during the mass screening of 100,000 RefSeq genomes done by Walker et al. (21). However, no further analysis of this bacteriocin gene cluster was done.

Annotation of the nearby genomic region revealed a complete biosynthetic gene cluster for a lantibiotic. Downstream of the bacteriocin structural genes were two genes predicted to encode lantibiotic modifying enzymes (LanM1 and LanM2) of the LanC-like super family (CDD: cl04955). Located between the two LanM genes was a gene predicted to encode a LanT$_P$ enzyme, a peptidase domain-containing ABC transporter of the SunT family (CDD: cl26602). The SunT family of peptidase exporters removes leader peptides of the double-glycine type, a common cleavage motif for bacteriocin leaders. The gene cluster found in this strain (57-27) appeared to be arranged as two operons, as no obvious immunity genes were found on the same strand as the biosynthetic genes. However, two open reading frames (ORFs) approximately 1,200 bp upstream on the opposing strand were annotated with transport/immunity function by BAGEL4. Indeed, BLAST searches resulted in matches to lantibiotic immunity ABC transporters of the MutE/EpiE family (NCBI: WP_065541939.1, E-value $2e^{-14}$). The two ORFs were, therefore, named LanFE.

We cloned genes lanA1-M2 (excluding lanE-F) into the inducible expression vector pRMC2 (22) and transformed the resulting plasmid (pRMC2_Romsacin) into *S. aureus* RN4220 by electroporation. Expression of the bacteriocin cluster was induced by adding anhydrous tetracycline (0–2 µg/mL) to the growth media of overnight cultures or of RN4220 carrying pRMC2_Romsacin. We then spotted cell-free supernatant of the overnight culture (treated at 100°C before use) onto a lawn of *Lactococcus lactis* as described in the previous section. Clear zones were observed for RN4220 expressing pRMC2_Romsacin after induction with anhydrous tetracycline concentrations of 0.08–0.12 µg/mL, but not for the wild type (no plasmid) nor for uninduced RN4220 carrying pRMC2_Romsacin.

The presence of a complete lantibiotic biosynthesis gene cluster in *S. haemolyticus* 57-27 combined with the heat stability and protease sensitivity of the antimicrobial substance strongly suggested that the strain was producing this two-peptide lantibiotic which was responsible for the antimicrobial activity. This was confirmed by heterologous

**TABLE 1** Predicted bacteriocin gene cluster in *S. haemolyticus* 57-27 genome

| Gene | Predicted function | Size | Homologs (GenBank) |
|------|-------------------|------|--------------------|
| *lanF* | Immunity/transport | 257 aa | ATP-binding cassette domain-containing protein WP_070835451.1 |
| *lanE* | Immunity/transport | 232 aa | ABC transporter permease WP_070835449.1 |
| *lanA1* | Core peptide | 62 aa | Plantaricin C family lantibiotic WP_070835453.1 |
| *lanA2* | Core peptide | 67 aa | Class II lanthipeptide, LchA2/BrtA2 family WP_070835455.1 |
| *lanM1* | Modification | 860 aa | Type 2 lanthipeptide synthetase LanM WP_252689559.1 |
| *lanT$_P$* | Transport and maturation | 705 aa | Peptidase domain-containing ABC transporter WP_070835459.1 |
| *lanM2* | Modification | 858 aa | Lantibiotic modifying enzyme SUM61214.1 |

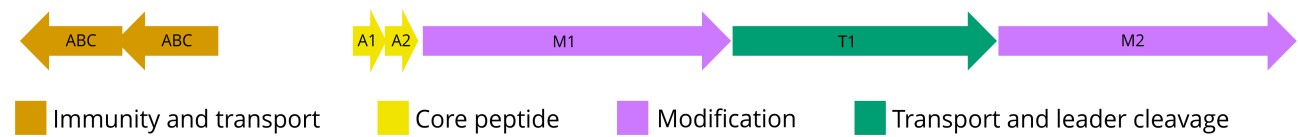

Immunity and transport          Core peptide          Modification          Transport and leader cleavage

**FIG 1** Bacteriocin encoding gene cluster in *S. haemolyticus* 57-27 genome. Adapted from BAGEL4.

expression of the bacteriocin cluster when induced with anhydrous tetracycline, in a different host, where it retained its ability to inhibit the *L. lactis* indicator strain.

## Bacteriocin purification

We purified the bacteriocin using a standard three-step scheme consisting of ammonium sulphate precipitation followed by cationic exchange and reversed-phase chromatography (RPC). The highest antimicrobial activity against *L. lactis* was found in RPC fractions with a concentration of around 25% 2-propanol, where we could see a peak in the RPC elution profile (indicated with an arrow) (Fig. 2). We used the fractions with the highest activity for further testing, indicated by the area with the darkest gray color in Fig. 2.

## Matrix-assisted laser desorption/ionization-time of flight mass spectrometry

Fractions showing antimicrobial activity were pooled and analyzed by matrix-assisted laser desorption/ionization-time of flight mass spectrometry (MALDI-TOF MS) to confirm the identity of the purified bacteriocin. The acquired MALDI-TOF MS spectra revealed the presence of two distinct peaks at 3,149.97 m/z and 3,548.16 m/z (Fig. 3). The two smaller peaks are likely the doubly charged ions of the same molecules (3,150/2 = 1,575, 3,548/2 = 1,774). To see if the two molecules correspond to the two-peptide lantibiotic (LanA1 and LanA2) found in the genome, we performed a structure prediction for the fully modified Lanα and Lanβ peptides to calculate their expected mass.

## Structure prediction

A prediction for the biosynthesis and final structures of two peptides was carried out based on the known modifications to the sequence-related lantibiotics lacticin 3147

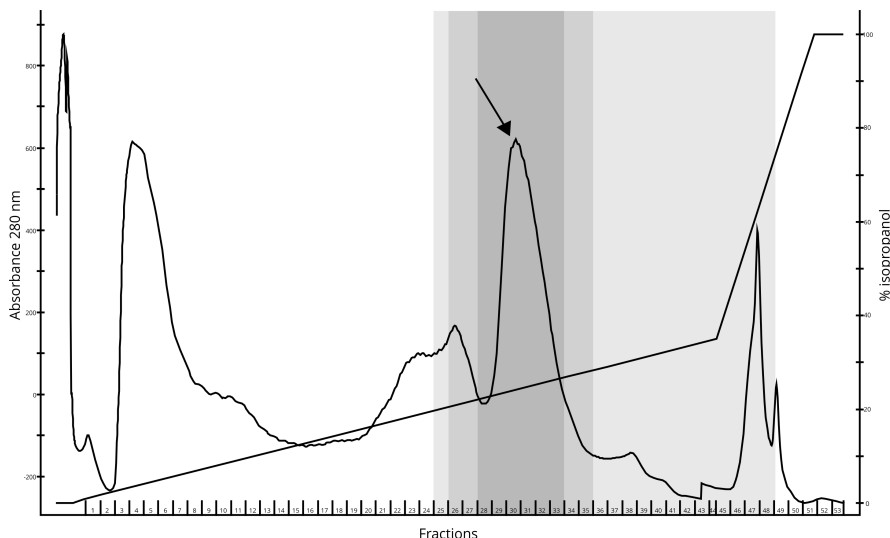

**FIG 2** Reversed-phase chromatography elution profile. Antimicrobial activity was the highest in fractions eluted at approximately 25% 2-propanol (containing 0.1% trifluoroacetic acid). The area with antimicrobial activity is colored gray (fractions 25–48). The area with darkest gray color has the highest antimicrobial activity (fractions 28–33), and the peak is indicated by an arrow. The fractions with the highest antimicrobial activity were pooled for further testing.

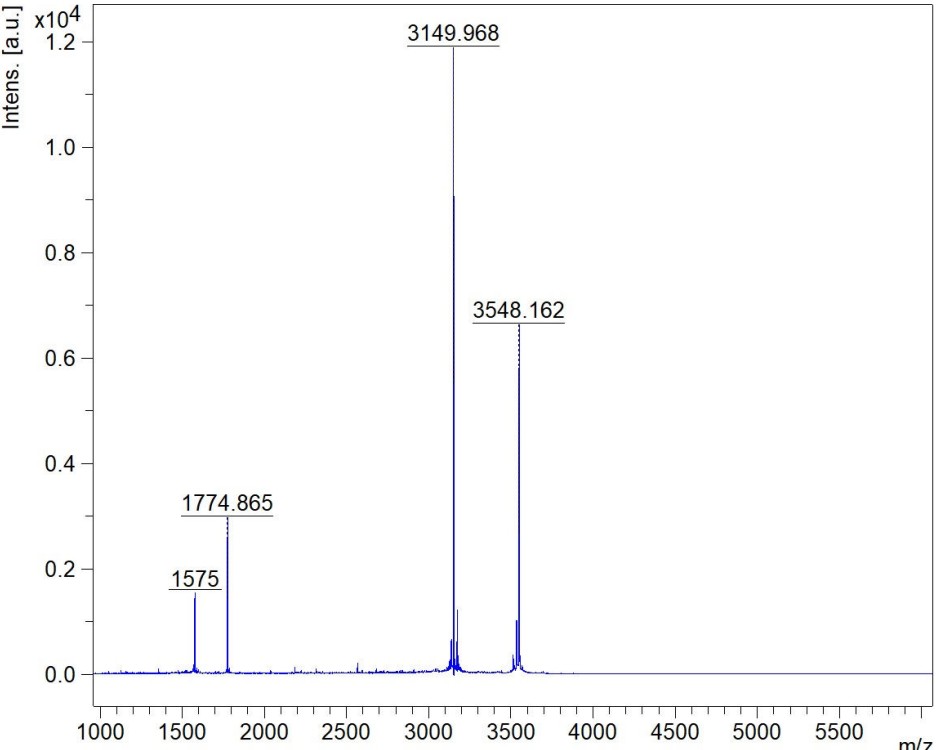

**FIG 3** MALDI-TOF MS analysis of pooled active fractions obtained after RPC.

and lichenicidin. Lichenicidin A1 and A2 core peptides share 40% and 44.7% sequence identity with the LanA1 and LanA2, respectively (23) (see Fig. 4).

Using the two-peptide lantibiotics lacticin 3147 and lichenicidin as templates for structure prediction, LanA1 was predicted to have three dehydrations ($-3 \times 18$ Da) and four reduced cysteines ($-4 \times 1$ Da). The peptide LanA2 was predicted to have nine dehydrations ($-9 \times 18$ Da) and four reduced cysteines ($-4 \times 1$ Da). A typical double-glycine leader was assumed for both peptides (see Fig. 4). The resulting theoretical monoisotopic mass of the predicted Lanα and Lanβ was 3,150.3 Da and 3,548.8 Da, respectively, which corresponded well with the masses obtained by MALDI-TOF MS (3,150.3–3,149.97 = 0.33 Da, 3,548.8–3,548.16 = 0.64 Da). The predicted biosynthetic scheme is presented in Fig. 5. After having identified a new bacteriocin, we have named the bacteriocin romsacin. Consequently, the lantibiotic structural peptides LanA1 and LanA2 were designated RomA1 and RomA2 (in unmodified form) and Romα and Romβ (in modified form).

## Bacteriocin antimicrobial activity

After obtaining purified romsacin, its antimicrobial spectrum against a range of Gram-negative and Gram-positive species was determined. Using a spot-on-lawn assay and planktonic growth, romsacin was shown to inhibit a broad range of Gram-positive species of both animal and human origin (Table S1; Table 2). Of potential clinical importance was the antimicrobial effect against several staphylococcal species and the WHO priority pathogens VRE and MRSA. The bacteriocin was also effective against the food-borne pathogens *Listeria monocytogenes* and *Bacillus cereus*. Gram-negative *Escherichia coli*, *Klebsiella pneumoniae*, *Acinetobacter baumannii*, and *Pseudomonas aeruginosa* strains were not inhibited by romsacin (Table S1).

*S. haemolyticus*, *S. epidermidis*, and *S. aureus* are often associated with biofilm-related infections from intravenous catheters, medical prostheses, and other implanted devices. For this reason, we wanted to see if romsacin was capable of disrupting biofilms formed

LanA1:

MSKLELLNESKANYLEKLTDEKIEETEAYGG|**KCSWWNASCHLGNNGKICTVSHECAAGCNL**

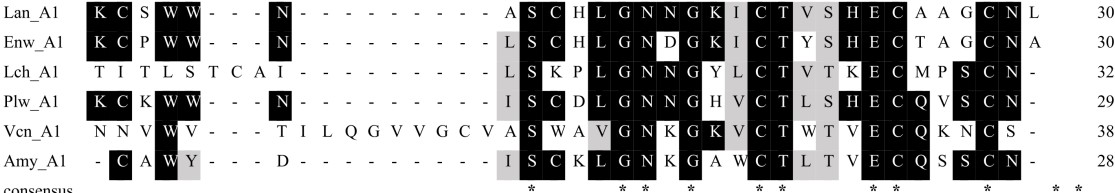

LanA2:

MFSKNFQRNEKMENTLKKVSSANDVNGG|**ATPTITTSSATCGGIIVAASAAQCPTLACSSRCGKRKK**

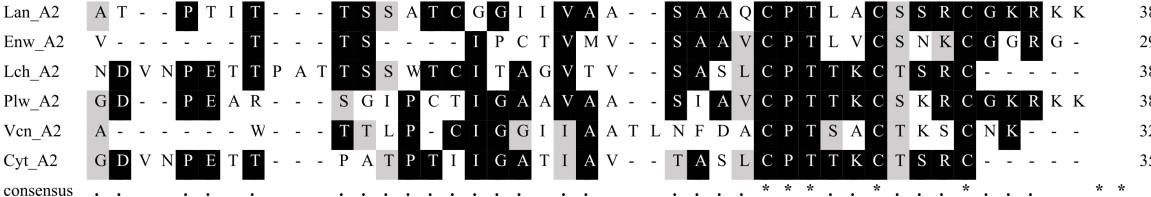

**FIG 4** Multiple-sequence alignment of LanA1 and LanA2 core peptides with the core peptides of other two-peptide lantibiotics. The precursor peptide sequences are shown above with the predicted cleavage site indicated with a bar ("|") and core peptide in bold. The sequence alignment was performed using T-Coffee. Aligned sequences are the core peptides of enterocin W (Enw; H3JSS9, H3JST0) produced by *Enterococcus faecalis*, lichenicidin (Lch; P86475, P86476) produced by *Bacillus licheniformis*, plantaricin W (Plw; D2KR94, Q9AF68) produced by *Lactiplantibacillus plantarum*, vagococcin T (URZ88908.1, URZ88906.1) produced by *Vagococcus fluvialis*, amyloliquecidin (Amy; CAG7845855.1) produced by *Bacillus velezensis*, and cytolysin (Cyt; KXO02964.1) produced by *Bacillus thuringiensis*.

by these species. We also wanted to test possible biofilm disruption of *E. faecium* biofilm. By using confocal microscopy together with a live (green) and dead (red) staining technique, we could show that romsacin appeared to effectively disrupt *S. epidermidis*, *S. haemolyticus*, MRSA, and VRE biofilms. As shown in Fig. 6, the number of green cells (live) was substantially reduced following treatment with romsacin compared to untreated controls.

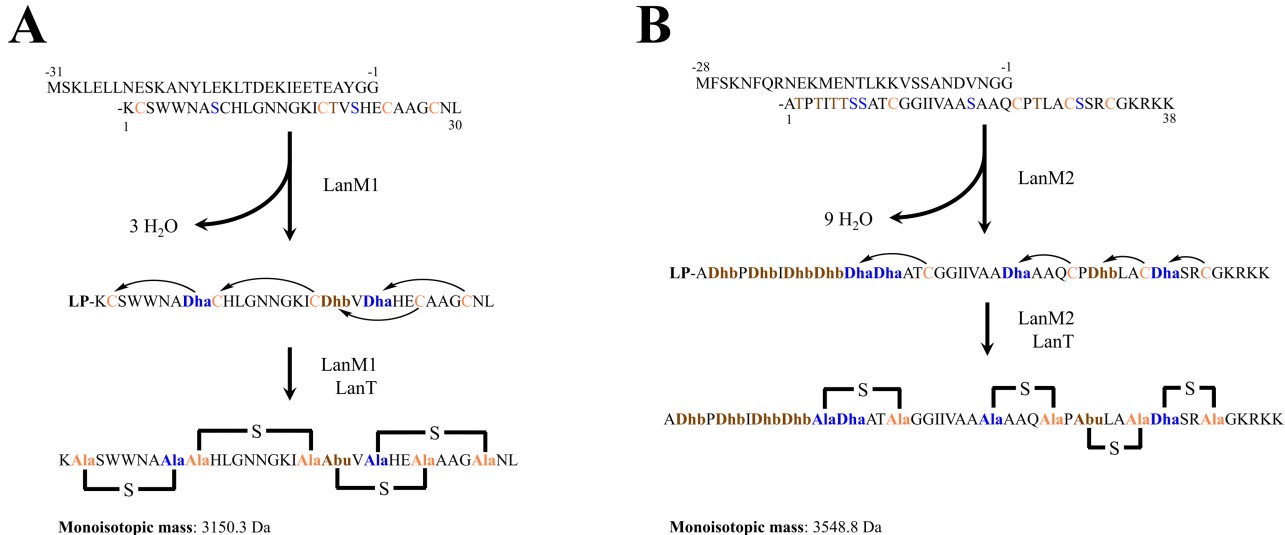

**FIG 5** Predicted post-translational modifications of peptide (A) RomA1 (to produce Romα) and (B) RomA2 (to produce Romβ) and their theoretical monoisotopic mass.

**TABLE 2** Romsacin inhibition against a panel of indicator strains growing on agar plates (spot-on-lawn assay) or planktonic[a]

| Number | Species | Agar inhibition | Planktonic inhibition, BU/mL |
|---|---|---|---|
| 1 | *Lactococcus lactis* 1403 control | +++ | Not tested |
| 1 | *Escherichia coli* | – | – |
| 1 | *Acinetobacter baumannii* | – | – |
| 1 | *Klebsiella pneumoniae* | – | – |
| 1 | *Enterococcus faecium* (VRE) | +++ | 47 |
| 2 | *Enterococcus faecium* (VRE) | ++ | 93 |
| 4 | *Enterococcus faecium* (VRE) | ++ | 47 |
| 6 | *Enterococcus faecium* | +++ | 47 |
| 10 | *Enterococcus faecium* | ++ | 93 |
| 1 | *Staphylococcus aureus* (MRSA) | ++ | 1493 |
| 3 | *Staphylococcus aureus* (MRSA) | ++ | 93 |
| 4 | *Staphylococcus aureus* (MRSA) | ++ | 747 |
| 5 | *Staphylococcus aureus* (MRSA) | ++ | 1493 |
| 7 | *Staphylococcus aureus* | + | 2987 |
| 10 | *Staphylococcus aureus* | ++ | 373 |
| 1 | *Staphylococcus haemolyticus* | ++ | 93 |
| 6 | *Staphylococcus haemolyticus* | ++ | 93 |
| 7 | *Staphylococcus haemolyticus* | +++ | 23 |
| 8 | *Staphylococcus haemolyticus* | ++ | 747 |
| 9 | *Staphylococcus haemolyticus* | ++ | 187 |
| 10 | *Staphylococcus haemolyticus* | ++ | 187 |
| 11 | *Staphylococcus haemolyticus* | ++ | 23 |
| 12 | *Staphylococcus haemolyticus* | ++ | 47 |
| 13 | *Staphylococcus haemolyticus* | Not tested | 47 |
| 14 | *Staphylococcus haemolyticus* | Not tested | 187 |
| 2 | *Staphylococcus lugdunensis* | + | 747 |
| 3 | *Staphylococcus lugdunensis* | ++ | 373 |
| 5 | *Staphylococcus lugdunensis* | ++ | 187 |
| 1 | *Staphylococcus saprophyticus* | ++ | 47 |
| 2 | *Staphylococcus saprophyticus* | +++ | 12 |
| 3 | *Staphylococcus saprophyticus* | ++ | 93 |
| 3 | *Staphylococcus epidermidis* | + | 1493 |
| 4 | *Staphylococcus epidermidis* | + | 373 |
| 6 | *Staphylococcus epidermidis* | Not tested | 747 |
| 1 | *Staphylococcus capitis* | – | 1493 |
| 3 | *Staphylococcus capitis* | ++ | 93 |
| 4 | *Staphylococcus capitis* | + | 187 |
| 2 | *Bacillus cereus* | ++ | 187 |
| 3 | *Bacillus cereus* | ++ | 747 |
| 14 | *Enterococcus faecalis* | ++ | 747 |
| 15 | *Enterococcus faecalis* | ++ | 1493 |
| 16 | *Enterococcus faecalis* | ++ | 747 |
| 39 | *Listeria monocytogenes* | ++ | 187 |
| 40 | *Listeria monocytogenes* | + | 373 |
| 63 | *Streptococcus uberis* | ++ | 93 |

[a]Purified romsacin (3 µL) spot-on-lawn assay; no zone (–), inhibition zone 1–6 mm (+), 7–12 mm (++), and ≥13 mm (+++). Inhibition of planktonic growth is shown as the highest dilution factor that inhibited the indicator by at least 50% compared to the control with no added antimicrobial.

## Pore formation assay

Propidium iodide (PI) is a fluorescent molecule where the fluorescence intensity (quantum yield) increases when intercalated in DNA. Intact bacterial cells are impermea-

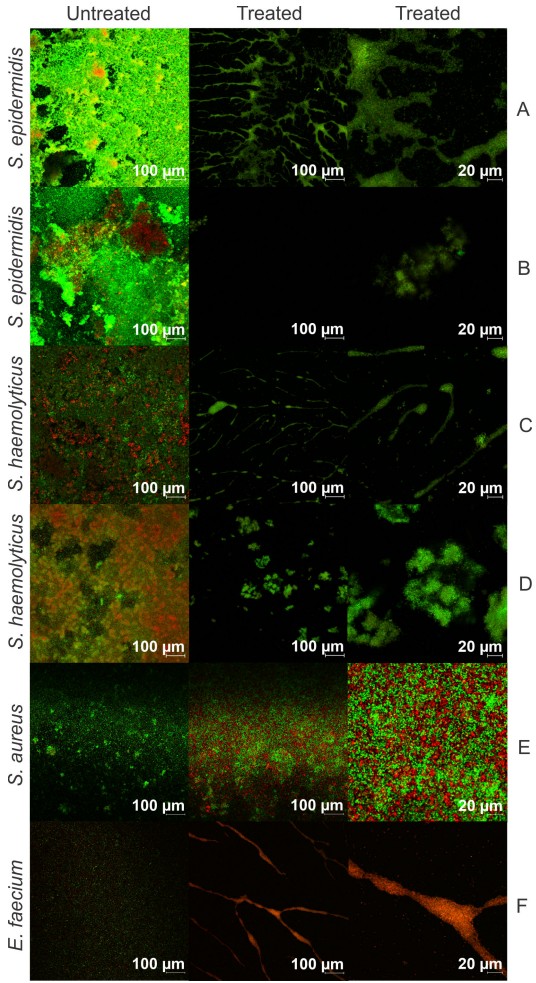

**FIG 6** Biofilm confocal microscopy of (A) *S. epidermidis* no. 4, (B) *S. epidermidis* no. 6, (C) *S. haemolyticus* no. 1, (D) *S. haemolyticus* no. 6, (E) *S. aureus* no. 1 (MRSA), and (F) *E. faecium* no. 2 (VRE). Column 1 shows untreated biofilms at 100× magnification. Columns 2 and 3 show biofilms after bacteriocin treatment with 100× and 400× magnification, respectively.

ble to PI, but the molecule will diffuse into cells with a damaged membrane, resulting in an increase in fluorescence. Cells treated with romsacin in the presence of PI showed very little increase in fluorescence, with values comparable to the negative control micrococcin P1, which do not affect membrane integrity. The pore-forming bacteriocin nisin A (positive control) showed a clear increase in fluorescence as expected. The results from the assay indicated that pore formation is unlikely to be the mode of action of romsacin against *L. lactis* (Fig. 7). As we could not determine the concentration of the bacteriocins used in the assay, all bacteriocins were tested at the same antimicrobial activity expressed in bacteriocin units (BUs). A BU was defined as the amount of bacteriocin that inhibited the indicator by 50% or more in 0.2 mL of culture.

## Scanning electron microscopy

The mode of action of most two-peptide lantibiotics characterized so far involves pore formation (24). As we could not see pore formation in *L. lactis* using the PI assay, we employed scanning electron microscopy (SEM) to confirm our results. Consistent with the PI assay, romsacin-treated *L. lactis* cells appeared intact (not lysed) but had a striated appearance which could not be seen in the untreated control (Fig. 8).

In order to investigate if mode of action is species dependent, we also performed SEM on MRSA, *S. haemolyticus*, *S. epidermidis*, and *Bacillus subtilis*. The integrity of

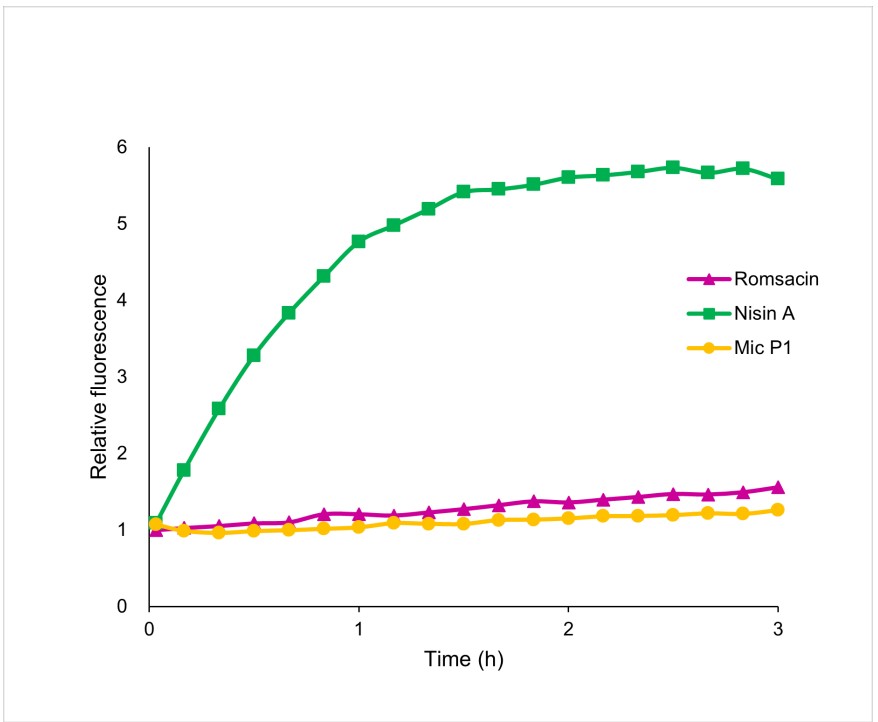

**FIG 7** Propidium iodide fluorescence over time (3 hours) combined with *L. lactis* IL1403 exposed to romsacin (purple), nisin (green), and micrococcin P1 (yellow). All bacteriocins were used at 50 BUs/mL.

staphylococcal cells did not seem affected after 30 minutes with romsacin treatment. For treated *B. subtilis* samples, we saw severely disrupted cells (Fig. 8).

The SEM analyses confirmed that bacterial lysis due to pore formation is not the mode of action for the novel bacteriocin in staphylococci and lactococci.

## Growth curves

The growth of both *S. haemolyticus* and MRSA treated with romsacin decreased markedly for around 2 hours (Fig. 9). After 2 hours, the growth of treated *S. haemolyticus* kept decreasing and was substantially reduced after 21 hours compared to the untreated growth control. For MRSA, the growth increased after 2 hours. The growth of non-treated MRSA increased throughout the experiment (Fig. 9).

The CFU assay was plated on agar within 1 hour after addition of bacteriocin or media to the cultures. At the start of the experiment, the CFU/mL for treated MRSA was $2.5 \times 10^7$, while for the untreated control, it was $6.3 \times 10^7$. Treated *S. haemolyticus* was $1.9 \times 10^4$ CFU/mL, while for the control, it was $5.2 \times 10^7$. The decrease in CFU, coupled with the rapid drop in optical density observed after the addition of the bacteriocin, indicates bacteriolytic effect against the majority of the *S. haemolyticus* cells. After 21 hours, CFU/mL for treated MRSA was $7.3 \times 10^8$, while for the untreated control, it was $9.3 \times 10^8$. Treated *S. haemolyticus* had 130 CFU/mL (small colony variants), while for the control, it was $1.1 \times 10^8$.

## Membrane integrity assay

We investigated the romsacin effect on membrane integrity by using a *B. subtilis* strain carrying a plasmid where luciferase is constitutively expressed. If romsacin affects the permeability of the cell, D-luciferin will enter the cell, and luminescence will be emitted. ATP is needed for light to be emitted. If the cell dies, there will be a strong drop of luminescence due to lack of ATP.

Untreated          Treated

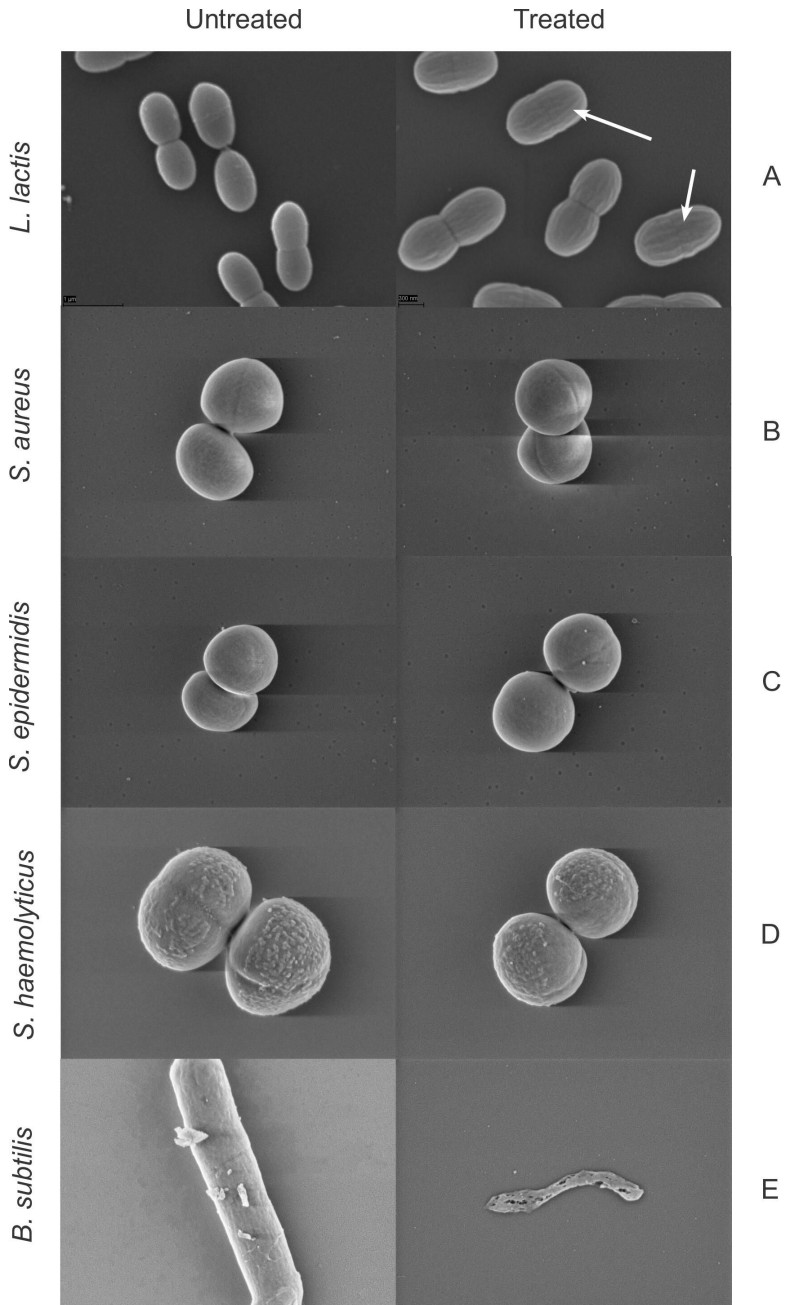

FIG 8 Scanning electron microcopy of (A) *L. lactis* IL1403, (B) *S. aureus* no. 1 (MRSA), (C) *S. epidermidis* no. 6, (D) *S. haemolyticus* no. 1, and (E) *B. subtilis* 168. All cells were exposed to bacteriocin for 30 minutes. Treated *L. lactis* cells (70,000× magnification) had a striated appearance (white arrows). The untreated *L. lactis* control is shown with a 50,000× magnification, and the staphylococci and *B. subtilis* are shown with a 40,000× magnification.

Romsacin had a quick rise in luminescence in the four first dilutions (Fig. 10), corresponding to the dilutions used in the MIC assay for *B. subtilis* (data not shown). The rise in luminescence was followed by a drop, indicating cell death. There was a clear difference in luminescence when comparing romsacin with chlorhexidine, which is known for its membrane disruptive properties (25). Chlorhexidine seems to affect the membrane faster than romsacin, as the drop in luminescence after treatment with chlorhexidine is observed immediately. For romsacin, there is a slower diffusion of D-luciferin, and it does not kill all cells during the four initial minutes. However, after

having completed all the 4-minute reads, we continued to monitor the luminescence for 10 hours to look at the long-term effect of romsacin (data not shown). At the start of the long-run experiment (within 1 hour after addition of romsacin), the relative luminescence units had dropped below 100 in the well with the most concentrated romsacin (1/20 dilution), indicating cell death.

## DISCUSSION

We have identified a new bacteriocin, romsacin, produced by *S. haemolyticus*, with relatively broad antimicrobial activity. The activity was confirmed by heterologous expression of the bacteriocin gene cluster in a different host. Two-peptide lantibiotics have previously been described in staphylococci (26, 27), but we believe this is the first description of a two-peptide lantibiotic in *S. haemolyticus*. The bacteriocin romsacin is active against a broad range of Gram-positive bacteria, including the WHO priority pathogens MRSA and VRE. The pathogens on the WHO priority list have been reported as a global health threat where we urgently need new antimicrobial treatment options (6). Several reports describe bacteriocins effective against MRSA and VRE (7, 28–30). Romsacin belongs to the lanthipeptides. Some, but not all, bacteriocins within that group are effective against MRSA (7). As different clinical strains have different resistance profiles, it is important to map out several possible therapeutic alternatives.

CoNS is part of the microbiota of skin and mucous membranes of humans and animals, and production of bacteriocins by CoNS is well known. However, the biological role of bacteriocins in host colonizers is not known, but findings suggest that bacteriocins promote host colonization by eliminating competitors (31–33). Several staphylococcal species produce bacteriocins, named staphylococcins, where the majority are classified as lantibiotics (34, 35). Six well-characterized bacteriocins have been described for *S. epidermidis*, and several staphylococcins have been shown to exert inhibitory activity against *S. aureus* and have a potential as treatment option to staphylococcal or other Gram-positive bacterial infections (34). Bacteriocin production by staphylococcal species inhabiting the human nose showed activity against several bacterial species in the nasal microbiota, such as *Moraxella catarrhalis* (36). A few publications describe bacteriocin production in *S. haemolyticus* from animal origin (7–9). One of the studies describes a *S. haemolyticus* bacteriocin with activity against a mastitis-related *S. aureus* strain (9). Romsacin is the first description of a bacteriocin from a commensal *S. haemolyticus* isolated from humans.

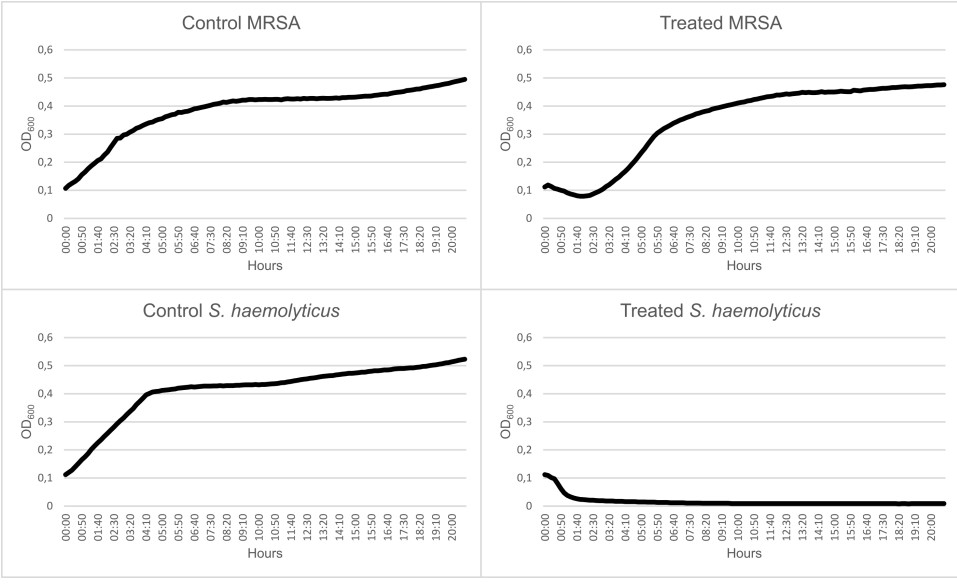

**FIG 9** Growth curve 0–21 hours of *S. aureus* MRSA (no. 1) and *S. haemolyticus* (no. 1) untreated or treated with romsacin.

Romsacin had no effect against *E. coli*, *A. baumannii*, or *K. pneumoniae*, as bacteriocins originating from Gram-positive bacteria are usually not effective against Gram-negative bacteria. However, some studies report that bacteriocins from Gram-positive bacteria can gain activity and act synergistically with other compounds known to inhibit growth or permeabilize the outer membrane of Gram-negative bacteria (37, 38). Nisin has been shown to be active against *E. coli* (39) and *Pseudomonas aeruginosa* when combined with outer membrane permeabilizer polymyxin B nonapeptide (PMBN) or metal ion chelator EDTA (40, 41). Similarly, the spectrum of activity of romsacin could potentially be expanded to include Gram-negative bacteria if used in combination with other compounds such as PMBN and EDTA. However, this remains to be investigated.

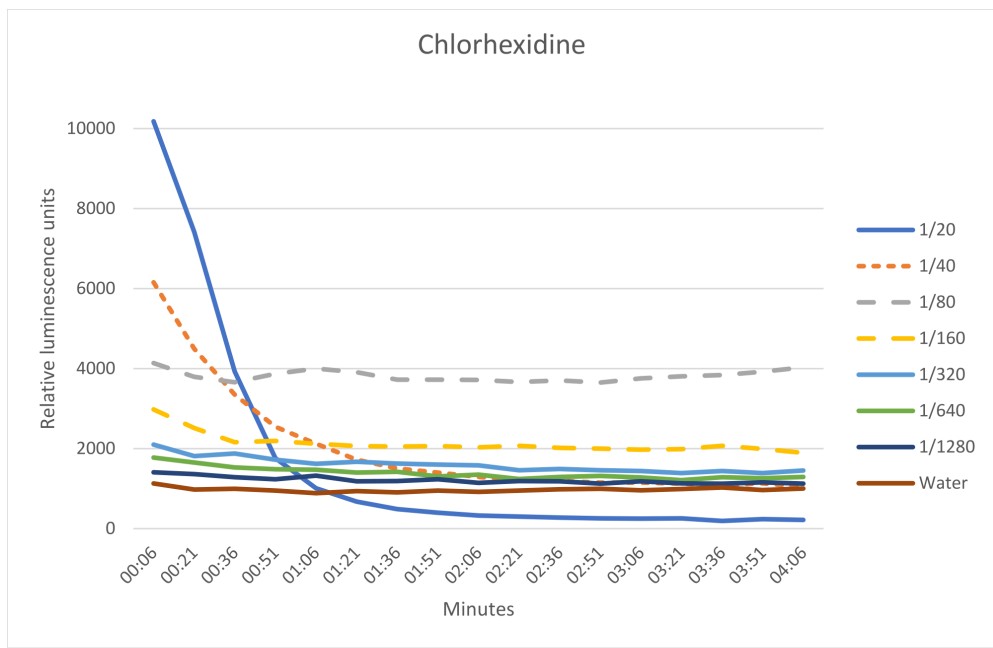

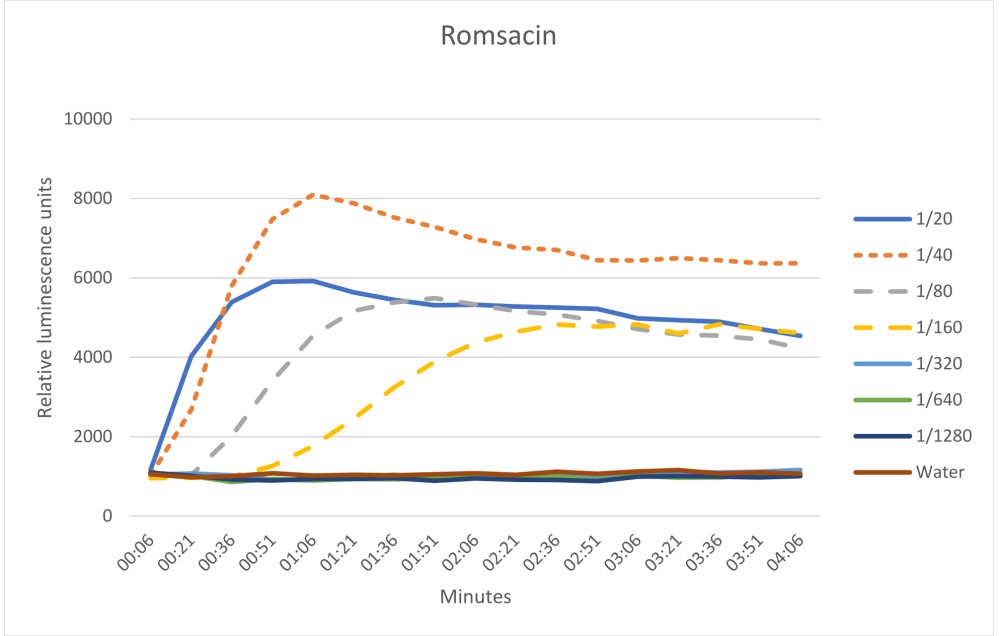

**FIG 10** Membrane integrity assay with *B. subtilis* 168 carrying the pCSS962 plasmid. The bacteria were treated with either chlorhexidine or romsacin, and luminescence was measured for 4 minutes. Seven dilutions of the antimicrobial compound were used (1/20 to 1/1,280) in addition to water. Readings were made 0–4 minutes after addition of chlorhexidine or romsacin.

Romsacin effectively eradicated the *S. epidermidis*, *S. haemolyticus*, MRSA, and VRE biofilms. Biofilm formation is a major virulence factor among staphylococci and enterococci, causing infections associated with foreign body surfaces, especially affecting patients with weakened immune systems (31, 32, 42–44). Microbial cells in biofilms are less susceptible to antibiotics than planktonic cells, caused by reduced metabolism and impaired diffusion/penetration of antibiotics (31, 43, 45, 46). Romsacin was shown to effectively disrupt both *S. haemolyticus*, *S. epidermidis*, MRSA, and VRE biofilms. However, fluorescent signals in treated samples of *S. epidermidis*, *S. haemolyticus*, and *E. faecium* were low, indicating a loss of biofilm/bacteria following treatment. The loss of biofilm was not of the same extent in the romsacin-treated *S. aureus* sample, but the number of live cells was markedly reduced compared to the control. Bacteria that have formed biofilms often have 10 to 1,000 times higher tolerance to antibiotics compared to planktonic cells (39). The bacteriocin gallidermin produced by *Staphylococcus gallinarum*, efficiently eradicated biofilms formed by *S. epidermidis* and *S. aureus* (47). Different bacteriocins have been shown to have various antibiofilm strategies, making them attractive candidates for biofilm eradication (48). As there are few effective treatment options against biofilms, new additions, such as romsacin, are needed.

Bacteriocins produced by staphylococci are commonly encoded on plasmids or other mobile genetic elements such as transposons but can also be chromosomally encoded (34). Lantibiotic gene clusters acquired by horizontal gene transfer have previously been described in *S. haemolyticus* strains originating from rice seeds (49). The prevalence of bacteriocin gene clusters on mobile genetic elements could suggest that they provide a benefit to their host. The romsacin gene cluster is located on a contig which has features indicating that it is part of a plasmid. Downstream of the romsacin gene cluster is a *repA* gene which initiates replication of plasmids. Also located in the same genomic region is a Tn552 DNA invertase gene and an IS6 family transposase, suggesting that the bacteriocin is likely part of a mobile genetic element.

The structure of romsacin was not determined experimentally with much certainty (by, e.g., MS/MS or crystal structure). However, lantibiotics that bind to lipid II contain a conserved lipid II binding motif GxxxTx(S/T)x(E/D)C (50). The (methyl)lanthionine ring structures form a defined binding pocket for lipid II and are, therefore, relatively predictable (51); the same motif is present in RomA1. This leaves few options for the remaining cysteines and serines/threonines (Ser/Thr). Although a varying number of Ser/Thr can remain unmodified in the final structure, the mass difference of 18 Da (corresponding to water) will correspond to the number of modified Ser/Thr. The β-peptide of two-peptide lantibiotics show much less homology to each other than the α-peptides, but many have a CPTxxCxxxC motif at the C-terminal end (52). Mutations introduced to alter the ring structures of the β-peptide of lacticin 3147 were inactive or not processed by the cognate LanM (53). This suggests that the ring structures of the β-peptides are also well conserved, despite much less being known about their role/function. By applying modifications consistent with lantibiotics to the two predicted lantibiotic precursors found in the genome, we obtained expected masses that almost exactly matched those obtained by MALDI-TOF MS. Taken together, we are confident the purified bacteriocin is derived from *romA1* and *romA2*.

Most lantibiotics have been shown to bind the cell wall synthesis precursor molecule lipid II. Among the single-peptide lantibiotics, two different but overlapping modes of action have been described (24). The type-A(I) lantibiotics such as nisin first interact with lipid II, thereby disrupting cell wall synthesis, but will subsequently insert into the membrane and aggregate into a pore complex (24). Nisin exposure causes leakage of intracellular contents (54). Lantibiotics of type-A(II) and type-B have not been shown to form pores but kill target cells by inhibition of cell wall synthesis and likely additional unknown factors (24). Two-peptide lantibiotics are believed to use the dual mode of action only, where the α-peptide forms a complex with lipid II which recruits the β-peptide to form a pore (23). The propidium iodide pore formation assay has been used previously to examine the mode of action of bacteriocins, including two-peptide

lantibiotics (55, 56). The mode of action of the bacteriocin vagococcin T, with sequence homology to romsacin (Fig. 4), is by forming pores in the bacterial cell membrane (55). However, we were not able to measure any pore formation in *L. lactis* using this assay. It could be that romsacin forms pores too small for the passage of PI and/or DNA but still permits the diffusion of essential ions such as $H^+$, $K^+$, and $PO_4^{3-}$, which leads to loss of turgor pressure. SEM micrographs of *L. lactis* showed cells of normal morphology, except all cells showed striations (lines) on the surface perpendicular with the septum that were not present in the control. The underlying peptidoglycan architecture of *L. lactis* is parallel to the septal plane, opposite of the striations (57). The striated appearance is likely a consequence of cell wall inhibition; however, we have not been able to explain its cause or structure. SEM micrographs of *S. aureus*, *S. epidermidis*, and *S. haemolyticus* also showed cells with normal morphology. Increased incubation time could have given other results and should be tested in the future. For *B. subtilis,* massive cell disruption was observed, which correlates well with the membrane integrity assay, where the romsacin-treated *B. subtilis* reporter strain showed rapid membrane leakage. Growth curves of romsacin-treated *S. haemolyticus* and *S. aureus* cells showed a rapid antimicrobial effect within 2 hours. This indicates that the bacteriocin has a bacteriolytic effect (58, 59). After 2 hours, the *S. aureus* cells regain growth, which displays single-cell resistance against romsacin, which can be explained by a heterogenous population (58). The confocal images of the *S. aureus* biofilms also showed that not all cells in the biofilm were eradicated to the same extent as it was observed for *S. haemolyticus* and *E. faecium*, supporting the single-cell resistance observed also in the growth curve. Combination treatment using romsacin and a second antimicrobial agent should, therefore, be tested in the future.

## Conclusion

In this study, we describe a new bacteriocin, romsacin, found in a commensal *S. haemolyticus* isolate. The bacteriocin has broad antimicrobial activity, both against planktonic cells and bacterial biofilms. Romsacin is a promising contributor to combat antibiotic-resistant pathogens. Further work is needed to establish the therapeutic potential of romsacin, both alone and in combinations with other compounds, and to determine its structure and mechanism of action.

## MATERIALS AND METHODS

### Detecting bacteriocin-producing *S. haemolyticus*

We screened overnight cultures from 174 *S. haemolyticus* isolates for bacteriocin inhibitory activity against three indicators: *Lactococcus lactis* IL1403 (60), a clinical *S. haemolyticus* 51-21 isolate (11, 19), and *Staphylococcus aureus* ATCC 25923. Colonies were picked from each of the 174 *S. haemolyticus* isolates from blood agar plates (Thermo Fisher Scientific, USA), then transferred to tryptic soy broth (TSB) (BD, USA/ Merck, Germany) and incubated with shaking at 37°C overnight.

We prepared 0.5 McFarland solutions in 0.85% saline of colonies from each of the indicator strains.

The suspensions were inoculated on Mueller Hinton (MH) agar (Oxoid, England) with a cotton swab and a rotator. Five microliters of overnight cultures, cell-free supernatant, or treated supernatant (heat, pH, protease) were spotted on the plates. Inhibition of bacterial growth was assessed visually after 20–24 hours. Three technical replicates were made of each plate. The genomes of *S. haemolyticus* isolates were submitted to the BAGEL4 webserver for identification of bacteriocin genes (20).

All except two *S. haemolyticus* isolates used in this study had been obtained and sequenced as part of previous studies (11, 19, 61). Of the isolates, 123 were of clinical origin, 46 were commensal isolates, and 4 were of veterinary origin. In addition, we tested a *S. haemolyticus*-type strain (CCUG 7323T) (62).

TABLE 3   Primer sets used for amplification of the bacteriocin cluster genes from *S. haemolyticus* 57-27 and plasmid pRMC2

| Primer | Set | Sequence 5'–3' | Extension | Product |
|--------|-----|----------------|-----------|---------|
| pRMC2_A1_FW | 1 | gtaccgttaggagggggttatttatgagtaaattagaactacttaatgaa | 65°C, 6:00 | 7,786 bp |
| pRMC2_A1_RV | | tgaattcgagctttatatgaataaactttctgagttggatgaaataag | | |
| pRMC2_A1_vec_FW | 2 | cccctcctaacgctaccatcatgcttattttaattatactctatcaatgatag | 3:30 | 6,439 bp |
| pRMC2_A1_vec_RV | | tttattcatataaagctcgaattcactggc | | |
| M2_INS_RV | 3 | gatgagatggaaggagatattattaatggaagtatagg | 1:00 | 785 bp |
| pRMC2_INS_FW | | gcctcttcgctattacgccag | | |
| M1_INS_FW | 4 | ccttcattatgactatcaccttggtttaattctatag | 1:00 | 1,084 bp |
| pRMC2_A1_vec_RV | | ctgttaatcactttacttttatctaatctagacatcattaattc | | |

## Heterologous expression of bacteriocin gene cluster

The genes required for bacteriocin core peptide production and those for modification, transport, and maturation were cloned into plasmid pRMC2 (Addgene, #68940) (Fig. S1). This plasmid allows anhydrous tetracycline-inducible expression of cloned genes (22).

We amplified the genes Lan A1-M2 (excluding Lan E-F) using primer set 1 (Table 3), following a two-step PCR protocol due to the AT-rich nature of the bacteriocin gene cluster sequence (63). We amplified the pRMC2 plasmid by PCR using primer set 2 (see Table 3 below). Both PCRs used Q5 High-Fidelity 2× Master Mix [New England Biolabs (NEB), USA]. Amplicons from both PCRs were digested with DpnI (NEB) before being cleaned up using the E.Z.N.A. Cycle Pure Kit (Omega, USA). We assembled the amplicons using NEBuilder HiFi DNA Assembly Master Mix (NEB) to form plasmid pRMC2_Romsacin. The newly assembled plasmids were transformed into NEB 5-alpha Competent *E. coli*, which we spread out onto Luria-Bertani (LB) + 100 µg/mL ampicillin and incubated overnight at 37°C. Correct assembly of the bacteriocin cluster in the plasmid was confirmed by colony PCR using primer sets 3 and 4 and OneTaq 2× MasterMix (NEB). We isolated the plasmids from *E. coli* using the NucleoSpin Plasmid Kit (Macherey Nagel, Germany) and concentrated them using Pellet Paint (Merck, USA).

We selected *S. aureus* RN4220 as a host for heterologous gene expression due to the ease with which it can be transformed, compared with other staphylococci. To make competent RN4220, we grew an overnight culture in 5 mL of TSB (37°C, shaking at 250 rpm) and diluted it with pre-warmed TSB to an optical density of 0.5 at 600 nm. The bacteria were returned to the incubator for 40 minutes before being harvested by centrifuging at 5,000 × *g* for 10 minutes. The pellet was washed in ice-cold sterile Milli-Q water before centrifuging at 5,000 × *g*. This step was repeated once. Following washing, we resuspended the cells in a 1:10 volume of ice-cold sterile 10% glycerol before centrifuging at 5,000 × *g* for 10 minutes. This step was repeated, but the volume of 10% glycerol was successively reduced each subsequent step to 1:25, 1:10, 1:100, and finally 1:200. Competent cells were aliquoted and frozen at −70°C until use.

Before electroporation, the competent cells were thawed on ice for 5 minutes and then on the bench for 5 minutes before being centrifuged at 5,000 × *g* for 1 minute. The supernatant was removed, and the cells were resuspended in sterile 10% glycerol with 0.5 M sucrose. We added 1 µg of plasmid to the cells and incubated them on the bench for 10 minutes. The cells were then transferred to a 1-mm electroporation cuvette (Biorad) and electroporated at 2.5 kV, 100 Ω, 25 µF (GenePulser Xcell, Biorad). We added 950 µL of TSB + 0.5 M sucrose (filter sterilized) to the cells and transferred them to a clean Eppendorf tube before incubating them for 1 hour at 37°C with shaking at 250 rpm. After recovery, we plated out 100-µL aliquots onto TSB + 10 µg/mL chloramphenicol before overnight incubation at 37°C. Presence of the plasmid was confirmed by PCR.

To induce the expression of the gene cluster, we added anhydrous tetracycline (0–2 µg/mL) to the TSB growth media of overnight cultures of RN4220 carrying pRMC2_Romsacin. We spotted 5 µL of cell-free supernatant (treated at 100°C before use) on plates of *L. lactis* IL 1403 indicator strain, as described in the previous section. As controls, we used wild-type RN4220 (no plasmid) and growth media with

anhydrous tetracycline (no bacteria). We used the *S. haemolyticus* bacteriocin producer for comparison of the results.

## Bacteriocin stability

We exposed aliquots of concentrated cell-free supernatants to various treatments prior to antimicrobial testing, performed as described above. The aliquots were exposed to 4, 10, 20, 30, 40, 50, 80, 90, 100, or 121°C for 15 minutes. The pH was adjusted to 2.1, 8.6, 9.3, 10.5, and 11.9 with sodium hydroxide (NaOH) or hydrochloric acid (HCl) and incubated at room temperature for 30 minutes. We used trypsin (200 µg/mL) to test protease sensitivity. Concentrated cell-free supernatant was treated with the enzyme for 1.5 hour at 37°C.

## Bacteriocin purification

Bacteriocin purification was performed similarly as described by Ovchinnikov et al. (56), with some modifications. One liter of BHI was inoculated with 2% (vol/vol) of an overnight culture of *S. haemolyticus* 57-27. The culture was incubated with vigorous shaking at 37°C for 24 hours, before cells were removed by centrifugation (10,000 × *g*, 4°C, 35 minutes). Proteins were then precipitated by the addition of 373-g ammonium sulphate per liter supernatant and left at 4°C overnight. Precipitated proteins were collected by centrifugation (12,000 × *g*, 4°C, 45 minutes). The protein pellet was dissolved in 200-mL Milli-Q water (Invitrogen, USA) and filtered through a 0.2-µm filter (Millipore, USA). The crude concentrate was freeze dried until use.

Freeze-dried concentrate precipitated from 1-L culture was dissolved in 200-mL Milli-Q water. The pH was adjusted to 4.5 (±0.5) and then applied on a HiPrep 16/10 SP-XL column (GE Healthcare, USA) equilibrated with Milli Q water (pH 4.5). The column was washed with 100 mL of 20 mM sodium phosphate buffer (pH 7) before elution of the bacteriocin with 100 mL of 0.5 M NaCl. The eluate was applied to a resource RPC column (1 mL) connected to an ÅKTA purifier system (GE Healthcare, USA). Water containing 0.1% trifluoroacetic acid (TFA) (Sigma-Aldrich, USA) was used as buffer A. We used a linear gradient of 2-propanol (Merck, USA) with 0.1% TFA (buffer B) for elution. The flow rate was 2–4 mL/min.

Antimicrobial activity in RPC purified fractions was determined quantitatively in 96-well plates using *L. lactis* 1403 as indicator strain. Briefly, overnight culture of *L. lactis* 1403 was diluted 50-fold in GM17 broth (Oxoid, England) in the wells of 96-well plates (Sarstedt, Germany) containing a serial dilution of the RPC fraction following incubation for 5–6 hours at 30°C. The growth was measured spectrophotometrically at 600 nm using SPECTROstarNano (BMG LABTECH, Germany). Purification was repeated so bacteriocin from 4 L of bacterial culture was purified all together. Fractions with bacteriocin activity were pooled.

## MALDI-TOF mass spectrometry

MALDI-TOF MS was performed on an ultrafleXtreme mass spectrometer (Bruker Daltonics, Bremen, Germany) in reflectron mode. The instrument was calibrated with peptide calibration standard II (Bruker Daltonics), and positive ions in the range 1,000 to 6,000 m/z were analyzed. The RPC purified fraction and matrix (HCCA; α-cyano-4-hydroxycinnamic acid) were mixed in equal volumes and spotted on a Bruker MTP 384 steel target plate (Bruker Daltonics) for analysis.

## Bacteriocin inhibition

The activity of the purified fractions was tested against WHO priority pathogens and a broad range of Gram-positive indicators with agar spot-on-lawn assay and planktonic growth inhibition (Table S1; Table 2).

We used a similar method as described by Holo (64) for the spot-on-lawn assy. Briefly, we made a 50-fold dilution of overnight culture of indicator strains in 5-mL BHI soft agar and plated out as a lawn on BHI agar plates (BD, USA). Afterwards, we spotted 3 µL of the bacteriocin on the lawn and incubated at 30°C for 24 hours. Inhibition of bacterial growth appeared as clear zones.

We performed planktonic growth inhibition by following the colony suspension (3A) and broth microdilution for antimicrobial peptides (4E) methods in the Wiegand protocol (65). The starting concentration of the bacteriocin in the MIC assay was a 1/10 or 1/5 dilution of the purified bacteriocin in water. We used 96-well plates (Falcon, USA) and MH broth (BD, USA) for the dilution series and performed three technical replicates. We report the dilution factor resulting in 50% inhibition of the indicator strain.

## Biofilm confocal microscopy

We assessed the bacteriocin effect on biofilm-associated *S. haemolyticus* (nos. 1 and 6), *S. epidermidis* (nos. 4 and 6), MRSA (no. 1), and VRE (no. 2) cells by confocal microscopy. Biofilms were established in four-well cover glass slides (Thermo Fisher Scientific, USA). Overnight cultures were diluted 1:10 in TSB with 1% glucose, and 500 µL was transferred to each well in the glass slides. Staphylococcal biofilms grew 24 hours and *E. faecium* for 48 hours at 37°C before the wells were washed twice with PBS (Sigma-Aldrich, USA). We dissolved and diluted the purified bacteriocin 1/2 in TSB with 1% glucose before addition to the biofilm. Five hundred microliters of bacteriocin or control (TSB with 1% glucose) were added to the wells and incubated for 24 hours at 37°C. Wells were carefully washed twice with PBS and stained for 20 minutes with LIVE/DEAD BacLight Bacterial Viability Kit (Thermo Fisher Scientific, US) (1-µL dye per milliliter PBS). Dye was removed, and 500-µL PBS was added to each well.

For confocal microscopy, we used a Zeiss LSM780 equipped with a 10×/0.45 M27 Plan Apochromat objective with digital zoom and ZEN v.2.3 software (ZEISS, Germany). We used the SmartSetup function in ZEN to adjust the channels. Pictures are 212.55 × 212.55 µm, with a pixel size of 255 nm. We took pictures from representative areas in the chamber wells. All photos are taken using the same settings.

## Bacteriocin units

The appropriate BU concentrations for the propidium iodide pore formation assay and scanning electron microscopy were determined by a microtiter plate assay. Briefly, twofold dilutions of purified romsacin, micrococcin P1, and nisin A in M17 medium supplemented with 0.5% glucose (GM17) were prepared in the wells of a microtiter plate to a volume of 100 µL per well. Each well was inoculated with 100 µL of a 25-fold diluted overnight culture of *L. lactis* IL1403 (50-fold final dilution). A bacteriocin unit was defined as the amount of bacteriocin that inhibited the indicator strain by at least 50% in 200-µL culture compared to the turbidity of a positive control with no added antimicrobial. Turbidity was measured spectrophotometrically at 600 nm using a SPECTROStar Nano microplate reader (BMG LABTECH, Germany).

## Propidium iodide pore formation assay

An overnight culture of the indicator strain *L. lactis* IL1403 was washed twice in PBS (5,000 × $g$, 5 minutes), and resuspended to an $OD_{600}$ of 3. We used a black microtiter plate to dilute romsacin, nisin A, and micrococcin P1 to 50 BU/mL in 100 µL of PBS containing 40 µM propidium iodide (see section above for bacteriocin units; BU). We added 100 µL of indicator to a final OD of 1.5 to each well containing diluted antimicrobial substance. Fluorescence was kinetically measured every 10 minutes for 3 hours with excitation at 535/20 nm (515–555 nm) and emission at 630/40 nm (590–670 nm) using a Hidex Sense microplate reader (Hidex, Finland).

## Scanning electron microscopy

*L. lactis* IL1403 was grown to mid-log phase (OD$_{600}$ ~0.5) and incubated with 50 BU/mL of romsacin for 30 minutes at 30°C (see section above for bacteriocin units). We used a culture with no bacteriocin added as control. After incubation, cells were harvested by centrifugation at 10,000 × *g* for 5 minutes, washed twice in PBS, and resuspended in fixing solution (1.25%, wt/vol, glutaraldehyde, 2%, wt/vol, formaldehyde, PBS) for overnight incubation at 4°C. Fixed cells were then washed three times in PBS and allowed to sediment/attach to poly-L-lysine-coated glass coverslips at 4°C for 1 hour. Attached cells were dehydrated with an increasing ethanol series (30%, 50%, 70%, 90%, and 96%, vol/vol) for 10 minutes each and finally washed four times in 100% ethanol. Cells were dried by critical-point drying using a CPD 030 critical point dryer (BAL-TEC, USA). Coverslips were sputter coated with palladium-gold using a Polaron Range sputter coater (Quorum Technologies, UK). Microscopy was performed on an EVO 50 EP scanning electron microscope (Zeiss, Germany) at 20 kV and a probe current of 15 pA. The SEM analysis was performed twice independently.

Preparations for SEM analysis of MRSA (no. 1), *S. haemolyticus* (no. 1), *S. epidermidis* (no. 6), and *B. subtilis* 168 were done in the same manner as for *L. lactis*, but with some exceptions. We used a Leica EM CPD 300 critical point dryer (Leica, Germany), a Polaron sputter coater SC7640 (Quorum Technologies, USA), and a Gemini SEM 300 scanning electron microscope (Zeiss, Germany). We used romsacin concentrations above MIC for the respective strains in the SEM assay.

## Growth curve

We investigated the bacteriostatic or bacteriolytic potential of romsacin by making growth curves of MRSA (no. 1) and *S. haemolyticus* (no. 1). Overnight cultures in MH broth were diluted 1:50 in fresh media and grown to OD$_{600}$ 0.5. A pellet of romsacin was dissolved in MH broth and mixed 1:1 with the bacterial culture. Bacterial culture mixed with 1:1 with MH broth was used as control. A 96-well microplate was incubated in Synergy H1 (Bio-Tek, USA) at 37°C for 21 hours, and the turbidity of the solutions was read at OD$_{600}$ every 10 minutes. We made a CFU count at 0 and 21 hours.

## Membrane integrity assay

We investigated the membrane disruptive properties of romsacin by using a bioluminescence-based assay described by Virta et al. (66). The method measured membrane permeabilization with D-luciferin as a substrate. D-luciferin hardly crosses biological membranes at neutral pH, but membranolytic agents allow it to enter the cell and emit light.

The test strain was *B. subtilis* 168 carrying plasmid pCSS962, which expresses luciferase and emits luminescence if externally added D-luciferin enters the bacterial cells after membrane disruption. We used chlorhexidine (200 µg/mL) as a reference. Chlorhexidine is known for its membrane disruption properties (25). *B. subtilis* 168 were grown overnight in MH medium with 5 µg/mL chloramphenicol. A dilution of the overnight culture was made in MH medium without antibiotics, and the culture was grown for around 4 hours. Undiluted antimicrobial compounds and six dilutions were used (1/2 to 1/64), and water, as control. Five microliters of the antimicrobial dilution series and water were mixed with 95 µL of an over-day culture of *B. subtilis* in black round-bottom 96-well plates (Nunc, Denmark). Plates were read immediately in a Synergy H1 reader (BioTek, USA). Monitoring of luminescence was done from 0 to 4 minutes after addition of the antimicrobial compound.

## ACKNOWLEDGMENTS

We are thankful to the Advanced Microscopy Core Facility for their resources and support, Ina Høiland at the Paediatric Research Group for technical assistance, Hans-Matti Blencke for his support and assistance, and the graphical services for their help

(UiT, The Arctic University of Norway). We also appreciate Astrid Wolden for assistance in figure layout. We thank the reviewers for their contributions to the development of the manuscript.

The study was supported by grants from the Northern Norway Regional Health Authority (HNF1344-17). The publication charge for this article was funded by a grant from the publication fund of UiT The Arctic University of Norway. T.F.O. is supported by a grant from the Research Council of Norway (project number 275190). The funders were not involved in the design, data collection, or interpretation of the results obtained in this study.

## AUTHOR AFFILIATIONS

[1]Department of Clinical Medicine, Faculty of Health Sciences, Research Group for Child and Adolescent Health, UiT The Arctic University of Norway, Tromsø, Norway

[2]Faculty of Chemistry, Biotechnology and Food Science, Norwegian University of Life Sciences (NMBU), Ås, Norway

## AUTHOR ORCIDs

Runa Wolden  http://orcid.org/0000-0001-9328-2286
Kirill V. Ovchinnikov  http://orcid.org/0000-0002-4767-8666
Jorunn Pauline Cavanagh  http://orcid.org/0000-0003-2058-1431

## FUNDING

| Funder | Grant(s) | Author(s) |
| --- | --- | --- |
| Helse Nord RHF (Northern Norway Regional Health Authority) | HNF1344-17 | Jorunn Pauline Cavanagh |
| Universitetet i Tromsø (UiT) | 1437977 | Runa Wolden |
| Norges Forskningsråd (Forskningsrådet) | 275190 | Thomas F. Oftedal |

## AUTHOR CONTRIBUTIONS

Runa Wolden, Conceptualization, Data curation, Formal analysis, Investigation, Methodology, Validation, Visualization, Writing – original draft, Writing – review and editing | Kirill V. Ovchinnikov, Conceptualization, Data curation, Formal analysis, Investigation, Methodology, Supervision, Validation, Visualization, Writing – original draft, Writing – review and editing | Hermoine J. Venter, Conceptualization, Data curation, Formal analysis, Investigation, Methodology, Validation, Visualization, Writing – original draft, Writing – review and editing | Thomas F. Oftedal, Conceptualization, Data curation, Formal analysis, Investigation, Methodology, Validation, Visualization, Writing – original draft, Writing – review and editing | Dzung B. Diep, Conceptualization, Data curation, Funding acquisition, Methodology, Project administration, Resources, Supervision, Writing – review and editing | Jorunn Pauline Cavanagh, Conceptualization, Data curation, Formal analysis, Funding acquisition, Investigation, Methodology, Project administration, Resources, Software, Supervision, Validation, Visualization, Writing – original draft, Writing – review and editing

## DATA AVAILABILITY

The whole-genome sequencing assembly for isolate *S. haemolyticus* 57-27 is available in the European Nucleotide Archive at the accession number GCA_903969855.

## ADDITIONAL FILES

The following material is available online.

## Supplemental Material

**Supplementary Figure 1 (Spectrum00869-23-s0001.pdf).** Plasmid pRMC2.
**Supplementary Table 1 (Spectrum00869-23-s0002.xlsx).** Romsacin inhibition.

## Open Peer Review

**PEER REVIEW HISTORY (review-history.pdf).** An accounting of the reviewer comments and feedback.

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
