## [Reviewer comments · Microbiology Spectrum]

Microbiology Spectrum

The novel bacteriocin romsacin from *Staphylococcus haemolyticus* inhibits Gram-positive WHO priority pathogens

Runa Wolden, Kirill Ovchinnikov, Hermoine Venter, Thomas Oftedal, Dzung Diep, and Jorunn Cavanagh

Corresponding Author(s): Jorunn Cavanagh, UiT Norges arktiske universitet Institutt for klinisk medisin

Review Timeline:

Submission Date:	March 3, 2023
Editorial Decision:	May 30, 2023
Revision Received:	September 22, 2023
Accepted:	September 24, 2023

Editor: Sacha Pidot

Reviewer(s): Disclosure of reviewer identity is with reference to reviewer comments included in decision letter(s). The following individuals involved in review of your submission have agreed to reveal their identity: Sylvie REBUFFAT (Reviewer #2)

Transaction Report:

DOI: <https://doi.org/10.1128/spectrum.00869-23>

May 30, 2023

Dr. Jorunn Pauline Cavanagh
UiT Norges arktiske universitet Institutt for klinisk medisin
Department of Clinical Medicine
Tromsø 9037
Norway

Re: Spectrum00869-23 (The novel bacteriocin romsacin from *Staphylococcus haemolyticus* inhibits Gram-positive WHO priority pathogens)

Dear Dr. Jorunn Pauline Cavanagh:

Thank you for submitting your manuscript to Microbiology Spectrum. The reviewers have assessed your manuscript as well written and having merit, but were concerned that some of the experiments appear relatively preliminary and will require further experimental work before being considered for publication.

Link Not Available

Sincerely,

Sacha Pidot

Journals Department
Reviewer comments:

Reviewer #1 (Comments for the Author):

In this manuscript, the authors describe studies where they screened out novel bacteriocin from *Staphylococcus haemolyticus* collection and successfully identified new bacteriocin named romsacin. The genetic element encoding rosamine synthesis and function was identified. The structure of romsacin was also determined by MALDI-TOF Mass spectrometry and structural prediction. Bacteriocin activity of romsacin was effective against broad range of Gram-positive bacterial species. Especially, romsacin was also effective against clinically important pathogenic bacteria including WHO priority pathogens such as

Staphylococcus aureus and Enterococcus faecium, and food-borne pathogens Listeria monocytogenes and Bacillus cereus. However, its mode of action remains unclear as PI staining assay and SEM observation did not tell obvious trait of romsacin action.

Overall, data carried out by a number of highly experienced techniques to investigate bacteriocin peptide so that the characteristic feature of romsacin was remarkably described. The manuscript is scientifically sound and will provide impact on this field. Meanwhile, I still believe that several improvements are required to ensure rationality and importance of author's claim.

1. I could not find figure legends for figure 1, 2 and 3. Please provide legends for these figures. It will be great help for readers.
2. The authors are using the term "WHO priority pathogens" in the title. However, only some of the experiments actually examined about WHO priority pathogens such as *S. aureus* or *E. faecium*. For example, biofilm disruption assay, PI staining test, and SEM observations should be performed using any WHO priority bacteria. Alternatively, excluding "WHO priority pathogens" from the title could be an option.
3. Lines 89-91: This sentence seems appropriate to be included in the result section. Had this prediction already been done by previous study? If so, please provide citation.
4. Line 93, 180-181: "MRSA" does not stand just for *S. aureus*, but for "Methicillin-resistant" *S. aureus* as the author described in line 60 and elsewhere. Similar for VRE. I think it is not necessary to specify every time, and only abbreviations such as "VRE" and "MRSA" can be used only once after the first definition at line 60.
5. Fig 6: I cannot see scientific reason not to perform the biofilm experiment with *S. aureus*. *S. aureus* and *E. faecium* can also form biofilm in laboratory condition. Test with *S. aureus* or *E. faecium* strain (ideally MRSA or VRE) will ensure that romsacin can be effective agent against WHO priority pathogen infection.
6. Fig 7 and 8: It is little bit confusing that pore forming assay and SEM observation were carried out only with *L. lactis*. Since Fig. 6 shows that romzacin destroys staphylococcal biofilms (suggesting lytic activity against these bacteria rather than only growth inhibition), the mechanism should be investigated using this staphylococcus, not other model bacteria. One cannot exclude possibility that romsacin can form pore on Staphylococcus cell membrane even if it does not pore form on *L. lactis*.
7. I am curious the mode of action of romsacin is bacteriostatic or bacteriolytic. To address this issue, the author could try some simple test where bacteria is exposed with romsacin in liquid phase and then measure changes of CFU or turbidity of culture over incubation time. If CFU or culture turbidity decreases over time, it suggests that romsacin has bacteriolytic activity. Also, Fig 4 displays several bacteriocins similar to romsacin. Are there any previous report about the mode of action of these bacteriocins? If there are literatures, the manuscript can include further discussion about the mode of action of romsacin.
8. Lines 222-223: Please describe a little bit more detail and examples about "bacteriocin effective against MRSA and VRE". Then, please compare them with romsacin, so that characteristic feature and possible advantage of romsacin highlights novelty and impactfulness of this manuscript.
9. Lines 244-247: Effect of PMBN or EDTA on romsacin activity against Gram-negative bacteria should be addressed here in this study.
10. Lines 286-287: To clarify relationship between purified romsacin and the genetic element, constructing of deletion mutant of romA1/romA2 would be better evidence.
11. Table 2: The list of same names for distinct strain rows is not very informative for reader. Please specify at least strain name or any unique identifier (for example, Freeze no. in Supplementary table 1 can be used) within species.

Reviewer #2 (Comments for the Author):

The paper from R. Wolden et al. describes the isolation, purification and partial characterization of an antimicrobial peptide (a bacteriocin) produced by a strain of *Staphylococcus haemolyticus* that was selected among 174 clinical and commensal isolates. This bacteriocin was named romsacin by the authors and it was identified as a two-peptide lantibiotic. The authors characterized the producing gene cluster, isolated romsacin using a classical purification protocol, and performed MALDI-TOF MS that identified the molecular mass of romsacin. From these data, they predicted that romsacin was a novel two-peptide lantibiotic. Romsacin was shown to inhibit a panel of Gram positive bacteria among which *S. haemolyticus*, and the WHO priority pathogens vancomycin-resistant *Enterococcus faecium* (VRE) and methicillin resistant *S. aureus* (MRSA). It also destroyed *S. epidermidis* and *S. haemolyticus* biofilms. In addition, the authors performed preliminary experiments to tackle the mechanism of action of romsacin, and concluded that the formation of pores could not explain the romsacin antibacterial activity. The paper is well written, the study could be interesting and could open interesting perspectives, but many aspects are very

preliminary. The points detailed below need being taken into account.

Specific comments

- 1- The study appears preliminary as regard both the structural characterization, and its spectrum of activity and mechanism of action. The MALDI-TOF measurements only allowed obtaining the molecular masses of the two components of romsacin RomA1, RomA2. Paragraph lines 164-173 is only speculative. The mechanism of action of RomA1-A2 was determined for a part on *S. haemolyticus* and *S. epidermidis* (confocal microscopy) and on *L. lactis* IL1403 (propidium iodide, SEM). Please justify the choice of three different strains for these experiments.
- 2- Perturbation of the biosynthesis of peptidoglycane is another mechanism of action demonstrated for other lantibiotics. The hypothesis should be tested here.
- 3- In Table 2 that shows the antibacterial spectrum of romsacin, the references and origin of the strains tested and the units for the MICs should be added (the unit is provided for propidium iodide and SEM only). As *L. lactis* was used for the mechanism of action, the activity on this strain should be added in the Table 2, while it only appears in Table S1.
- 4- Page 7, lines 151-173: it is desirable to separate sections MALDI-TOF and structure prediction in two different sections.
- 5- As the authors argue for the activity of romsacin against VRE, MRSA pathogens, a toxicity assay should be included in the paper (at least hemolysis assay) to further identify its potential.
- 6- Figure 2: antibacterial activity should be indicated on the figure; fraction numbers are too small to be read. Perhaps indicate only a few fractions in bigger letters...; time should appear on the figure
- 7- Legend to Figure 3 is too elusive: "of an active fraction"; specify which fraction
- 8- Legend to figure 4: the names of producing bacteria should be indicated
- 9- Literature references: The names of bacteria (genus and species) and if useful of other organism should be in italics (Refs 6, 8, 14,...) and the titles should not include systematic capital letters (ref 16).
- 10- The sentence turns used in the materials and methods "We screened overnight cultures; line 332 "we tested a *S. haemolyticus* type strain" etc should be avoided and changed to the impersonal grammatical turn.
- 11- Typing mistake: line 293 suppress "and"

Staff Comments:

Preparing Revision Guidelines

Please return the manuscript within 60 days; if you cannot complete the modification within this time period, please contact me. If you do not wish to modify the manuscript and prefer to submit it to another journal, please notify me of your decision immediately so that the manuscript may be formally withdrawn from consideration by Microbiology Spectrum.

Response to reviewers

Thank you both for your valuable input regarding our manuscript. We have responded to the majority of your comments. We sincerely believe that the revisions made based on your feedback have improved the quality of the manuscript. Please see our detailed answers below.

Reviewer #1 (Comments for the Author):

In this manuscript, the authors describe studies where they screened out novel bacteriocin from *Staphylococcus haemolyticus* collection and successfully identified new bacteriocin named romsacin. The genetic element encoding romsacin synthesis and function was identified. The structure of romsacin was also determined by MALDI-TOF Mass spectrometry and structural prediction. Bacteriocin activity of romsacin was effective against broad range of Gram-positive bacterial species. Especially, romsacin was also effective against clinically important pathogenic bacteria including WHO priority pathogens such as *Staphylococcus aureus* and *Enterococcus faecium*, and food-borne pathogens *Listeria monocytogenes* and *Bacillus cereus*. However, its mode of action remains unclear as PI staining assay and SEM observation did not tell obvious trait of romsacin action. Overall, data carried out by a number of highly experienced techniques to investigate bacteriocin peptide so that the characteristic feature of romsacin was remarkably described. The manuscript is scientifically sound and will provide impact on this field. Meanwhile, I still believe that several improvements are required to ensure rationality and importance of author's claim.

1. I could not find figure legends for figure 1, 2 and 3. Please provide legends for these figures. It will be great help for readers.

- Sorry for not making it clear enough. Legends for all figures are provided after the references. Please see line 813 to 858.

2. The authors are using the term "WHO priority pathogens" in the title. However, only some of the experiments actually examined about WHO priority pathogens such as *S. aureus* or *E. faecium*. For example, biofilm disruption assay, PI staining test, and SEM observations should be performed using any WHO priority bacteria. Alternatively, excluding "WHO priority pathogens" from the title could be an option.

- Thank you for your input. We have now included the WHO priority pathogens *Staphylococcus aureus* and *Enterococcus faecium* in the biofilm disruption assay, and *S. aureus* in SEM observations. Please also see answers under point 5 and 6 below.

3. Lines 89-91: This sentence seems appropriate to be included in the result section. Had this prediction already been done by previous study? If so, please provide citation.

- Thank you for the comment, we have edited the paragraph to improve clarification. This paragraph is a summary of what was done in this study. The genomes were sequenced previously, and the reference has now been provided. The gene cluster was discovered in this work and is further described in this paper. Please see lines 89-90.

4. Line 93, 180-181: "MRSA" does not stand just for *S. aureus*, but for "Methicillin-resistant" *S. aureus* as the author described in line 60 and elsewhere. Similar for VRE. I think it is not necessary to specify every time, and only abbreviations such as "VRE" and "MRSA" can be used only once after the first definition at line 60.

- We have now changed the manuscript according to your input. Changes are marked in yellow in the marked-up manuscript.

5. Fig 6: I cannot see scientific reason not to perform the biofilm experiment with *S. aureus*. *S. aureus* and *E. faecium* can also form biofilm in laboratory condition. Test with *S. aureus* or *E. faecium* strain (ideally MRSA or VRE) will ensure that romsacin can be effective agent against WHO priority pathogen infection.

- Thank you for your valuable feedback. We have now included biofilm producers of MRSA and VRE in the biofilm experiment. Please see updated Figure 6, and updated paragraphs in results, discussion and material and methods (marked in yellow in the marked-up manuscript).

6. Fig 7 and 8: It is little bit confusing that pore forming assay and SEM observation were carried out only with *L. lactis*. Since Fig. 6 shows that romsacin destroys staphylococcal biofilms (suggesting lytic activity against these bacteria rather than only growth inhibition), the mechanism should be investigated using this staphylococcus, not other model bacteria. One cannot exclude possibility that romsacin can form pore on *Staphylococcus* cell membrane even if it does not pore form on *L. lactis*.

- We have included MRSA, *S. haemolyticus*, *S. epidermidis* and a *B. subtilis* strain in the SEM observations. Please see updated method, results, Figure 8 and discussion (marked in yellow in the marked-up manuscript). The cell surface of the staphylococci appeared the same both with and without romsacin treatment. The *B. subtilis* strain showed a severe disruption of the cell. We still don't know the mechanism in staphylococci, and perhaps the results would have been different with bacteriocin treatment longer than 30 minutes. However, that is for further investigation.

7. I am curious the mode of action of romsacin is bacteriostatic or bacteriolytic. To address this issue, the author could try some simple test where bacteria is exposed with romsacin in liquid phase and then measure changes of CFU or turbidity of culture over incubation time. If CFU or culture turbidity decreases over time, it suggests that romsacin has bacteriolytic activity. Also, Fig 4 displays several bacteriocins similar to romsacin. Are there any previous report about the mode of action of these bacteriocins? If there are literatures, the manuscript can include further discussion about the mode of action of romsacin.

- Thank you for your feedback. We have made a growth curve of MRSA and *S. haemolyticus* where we have compared untreated and romsacin treated growth. Please see Figure 9, and updated method, results and discussion (marked in yellow in the marked-up manuscript).
- We have now added information about the mode of action of vagococcin mentioned in Figure 4, please see lines 354-356.

8. Lines 222-223: Please describe a little bit more detail and examples about "bacteriocin effective against MRSA and VRE". Then, please compare them with romsacin, so that characteristic feature and possible advantage of romsacin highlights novelty and impactfulness of this manuscript.

- Thank you for pointing this out. We have added further details about other bacteriocins effective against MRSA and VRE, please see lines 274-277. Romsacin has broad antimicrobial activity and are also effective against biofilm. All clinically important strains don't have the same resistant profiles, and it's important to map out several alternatives (including romsacin) for future treatment options.

9. Lines 244-247: Effect of PMBN or EDTA on romsacin activity against Gram-negative bacteria should be addressed here in this study.

- Investigating the synergistic properties of romsacin with other antimicrobial agents is of high clinical relevance. However, demonstrating the synergistic potential of romsacin would require inclusion of several agents and bacterial strains, which is outside the scope of this first descriptive paper. This is something we would like to pursue in a follow-up study.

10. Lines 286-287: To clarify relationship between purified romsacin and the genetic element, constructing of deletion mutant of romA1/romA2 would be better evidence.

- Thank you for your suggestion. Instead of creating a *S. haemolyticus* deletion mutant, we heterologously expressed the bacteriocin cluster by cloning the lanA-M2 genes into the inducible expression vector pRMC2 and transforming it into *S. aureus* RN4220. When induced, RN4220 carrying the plasmid was able to inhibit the growth of *Lactococcus lactis* as the original *S. haemolyticus* strain 57-27 was able to do. Neither the RN4220 wild type, nor uninduced RN4220+pRMC2_Romsacin, was able to produce clear zones in the *L. lactis* lawn. These additions can be found in the text in lines 136-143, 148-150, 268-269, and 404-449. We trust that this provides the supporting evidence for the activity of the bacteriocin cluster requested by the reviewer.

11. Table 2: The list of same names for distinct strain rows is not very informative for reader. Please specify at least strain name or any unique identifier (for example, Freeze no. in Supplementary table 1 can be used) within species.

- Thank you for your comment. We have now included more information in the table to guide the reader.

Reviewer #2 (Comments for the Author):

The paper from R. Wolden et al. describes the isolation, purification, and partial characterization of an antimicrobial peptide (a bacteriocin) produced by a strain of *Staphylococcus haemolyticus* that was selected among 174 clinical and commensal isolates. This bacteriocin was named romsacin by the authors and it was identified as a two-peptide lantibiotic. The authors characterized the producing gene cluster, isolated romsacin using a classical purification protocol, and performed MALDI-TOF MS that identified the molecular mass of romsacin. From these data, they predicted that romsacin was a novel two-peptide lantibiotic. Romsacin was shown to inhibit a panel of Gram-positive bacteria among which *S. haemolyticus*, and the WHO priority pathogens vancomycin-resistant *Enterococcus faecium* (VRE) and methicillin resistant *S. aureus* (MRSA). It also destroyed *S. epidermidis* and *S. haemolyticus* biofilms. In addition, the authors performed preliminary experiments to tackle the mechanism of action of romsacin and concluded that the formation of pores could not explain the romsacin antibacterial activity.

The paper is well written, the study could be interesting and could open interesting perspectives, but many aspects are very preliminary. The points detailed below need being taken into account.

Specific comments

1- The study appears preliminary as regard both the structural characterization, and its spectrum of activity and mechanism of action. The MALDI-TOF measurements only allowed obtaining the molecular masses of the two components of romsacin RomA1, RomA2. Paragraph lines 164-173 is only speculative. The mechanism of action of RomA1-A2 was determined for a part on *S. haemolyticus* and *S. epidermidis* (confocal microscopy) and on *L. lactis* IL1403 (propidium iodide, SEM). Please justify the choice of three different strains for these experiments.

- Thank you for your feedback. We agree that some of our work is preliminary. However, this is a first descriptive paper, and we will include more assays in follow-up studies. We agree that the structure prediction described in lines 164-173 is speculative, as the sequence-related lantibiotics lacticin 3147 and lichenicidin were used. A crystal structure would provide further confirmation of structure, but this will be pursued in follow-up work.
- We chose *S. epidermidis* and *S. haemolyticus* strains because they are good biofilm producers and often cause implant-associated infections. *L. lactis* was chosen because it is commonly used as a model organism in the lab when testing bacteriocin mode of action. We have now also included MRSA and VRE (WHO priority list) in the biofilm confocal microscopy analysis. *S. haemolyticus*, *S. epidermidis*, and *S. aureus* are now also included in the SEM analysis.

2- Perturbation of the biosynthesis of peptidoglycane is another mechanism of action demonstrated for other lantibiotics. The hypothesis should be tested here.

- Thank you for your comment. To see if the cell surface is affected, we have chosen to do a membrane integrity assay of the model organism *B. subtilis*. Please see lines 248-264, 578-593, and Figure 10.

3- In Table 2 that shows the antibacterial spectrum of romsacin, the references and origin of the strains tested and the units for the MICs should be added (the unit is provided for propidium iodide and SEM only).

As *L. lactis* was used for the mechanism of action, the activity on this strain should be added in the Table 2, while it only appears in Table S1.

- Thank you for your comment. We have now added strain references and *L. lactis* in Table 2.
- We have added the BU/mL for the MIC units in Table 2 and Table S1.

4- Page 7, lines 151-173: it is desirable to separate sections MALDI-TOF and structure prediction in two different sections.

- Thank you for your suggestion. We have now separated the sections, please see line 159 and 168.

5- As the authors argue for the activity of romsacin against VRE, MRSA pathogens, a toxicity assay should be included in the paper (at least hemolysis assay) to further identify its potential.

- Investigating the toxicity is highly relevant for further application. However, we believe this can be included in later studies, and is outside the scope of this first descriptive paper.

6- Figure 2: antibacterial activity should be indicated on the figure; fraction numbers are too small to be read. Perhaps indicate only a few fractions in bigger letters...; time should appear on the figure.

- Thank you for your feedback. Due to lack of space in the figure, it was difficult to write the fractions in larger font. However, we have added shades of gray to make it easier to

understand in which of the fractions we find antimicrobial activity. The elution peak is indicated by an arrow.

- The RPC software has not generated a time curve, so it does unfortunately not appear on the figure. However, we have written in the method section that we used 200 ml in each purification, and that the flow rate was 2-4 mL/min.

7- Legend to Figure 3 is too elusive: "of an active fraction"; specify which fraction

- We have now changed the figure legend to «pooled active fractions». The fractions with the highest antimicrobial activity were pooled and used for further testing. The area for the pooled fractions is now also indicated on Figure 2.

8- Legend to figure 4: the names of producing bacteria should be indicated

- We have now indicated the names of the producing bacteria in the figure legend, lines 828-832.

9- Literature references: The names of bacteria (genus and species) and if useful of other organism should be in italics (Refs 6, 8, 14,...) and the titles should not include systematic capital letters (ref 16).

- Thank you for your comment, we have now corrected the references.

10- The sentence turns used in the materials and methods "We screened overnight cultures; line 332 "we tested a *S. haemolyticus* type strain" etc should be avoided and changed to the impersonal grammatical turn.

- Thank you for your comment. We believe that using impersonal or personal terms is subjective, so we choose to keep our style after consulting the journal regarding their guidelines and advice on this matter. You can find the journal's response below:

Dear Dr. Wolden,

Thank you for your interest in our journals. Either approach is fine: We screened, we tested or Cells were screened, strains were tested. The only caveat for the first approach is that if a

paper has only one author, "I" would be used instead of "we." (In that case, some authors prefer to use the impersonal approach rather than sprinkle "I" throughout the paper.)

We do not have style specific resources available, although our <https://journals.asm.org/editorial-style?journal=spectrum> page does list the references that underpin our style.

If you have other questions about the submission process, please let me know.

Becky Zwadyk

Spectrum

11- Typing mistake: line 293 suppress "and"

- Thank you for your observation. We have now corrected the typing mistake, line 349.

September 24, 2023

Dr. Jorunn Pauline Cavanagh
UiT Norges arktiske universitet Institutt for klinisk medisin
Department of Clinical Medicine
Tromsø 9037
Norway

Re: Spectrum00869-23R1 (The novel bacteriocin romsacin from *Staphylococcus haemolyticus* inhibits Gram-positive WHO priority pathogens)

Dear Dr. Jorunn Pauline Cavanagh:

Your manuscript has been accepted, and I am forwarding it to the ASM Journals Department for publication. You will be notified when your proofs are ready to be viewed.

Sincerely,

Sacha Pidot
Editor, Microbiology Spectrum
